# The Inhibitory Effects of Anti-ERC/Mesothelin Antibody 22A31 on Colorectal Adenocarcinoma Cells, within a Mouse Xenograft Model

**DOI:** 10.3390/cancers14092198

**Published:** 2022-04-28

**Authors:** Gentaro Taniguchi, Kazunori Kajino, Shuji Momose, Harumi Saeki, Liang Yue, Naomi Ohtsuji, Masaaki Abe, Tomoyoshi Shibuya, Akira Orimo, Akihito Nagahara, Sumio Watanabe, Okio Hino

**Affiliations:** 1Department of Molecular Pathology, Graduate School of Medicine, Juntendo University, 2-1-1 Hongo, Bunkyo-ku, Tokyo 113-8421, Japan; gtanigu@juntendo.ac.jp (G.T.); l-yue@juntendo.ac.jp (L.Y.); kadowaki@juntendo.ac.jp (N.O.); masa@juntendo.ac.jp (M.A.); aorimo@juntendo.ac.jp (A.O.); ohino@juntendo.ac.jp (O.H.); 2Department of Gastroenterology, Faculty of Medicine, Juntendo University, 2-1-1 Hongo, Bunkyo-ku, Tokyo 113-8421, Japan; tomoyosi@juntendo.ac.jp (T.S.); nagahara@juntendo.ac.jp (A.N.); sumio@juntendo.ac.jp (S.W.); 3Department of Human Pathology, Graduate School of Medicine, Juntendo University, 2-1-1 Hongo, Bunkyo-ku, Tokyo 113-8421, Japan; haru-s@juntendo.ac.jp; 4Department of Pathology, Saitama Medical Center, Saitama Medical University, 1981 Kamoda, Kawagoe 350-8550, Japan; momose@saitama-med.ac.jp

**Keywords:** colorectal cancer, mesothelin, ERC, molecular targeting therapy, anti-ERC antibody

## Abstract

**Simple Summary:**

The expression of Renal Carcinoma (ERC)/mesothelin is overexpressed in malignancies such as mesothelioma, pancreatic cancer, and ovarian cancer, and molecular-targeted therapies against ERC/mesothelin have been developed to treat them. Recently, it was revealed that ERC/mesothelin is also expressed in colorectal cancer; thus, this protein is expected to be a therapeutic target in colorectal cancer. In this study, we demonstrated that anti-ERC/mesothelin antibody 22A31 suppressed the growth of colorectal cancer cells subcutaneously xenografted on the back of mice. This is the first report to show the effectiveness of an anti-ERC/mesothelin antibody for the treatment of colorectal cancer in vivo.

**Abstract:**

The expression of Renal Carcinoma (ERC)/mesothelin is enhanced in a variety of cancers. ERC/mesothelin contributes to cancer progression by modulating cell signals that regulate proliferation and apoptosis. Based on such biological insights, ERC/mesothelin has become a molecular target for the treatment of mesothelioma, pancreatic cancer, and ovarian cancer. Recent studies revealed about 50–60% of colorectal adenocarcinomas also express ERC/mesothelin. Therefore, colorectal cancer can also be a potential target of the treatment using an anti-ERC/mesothelin antibody. We previously demonstrated an anti-tumor effect of anti-ERC antibody 22A31 against mesothelioma. In this study, we investigated the effect of 22A31 on a colorectal adenocarcinoma cell line, HCT116. The cells were xenografted into BALB/c nu/nu mice. All mice were randomly allocated to either an antibody treatment group with 22A31 or isotype-matched control IgG1κ. We compared the volume of subsequent tumors, and tumors were pathologically assessed by immunohistochemistry. Tumors treated with 22A31 were significantly smaller than those treated with IgG1κ and contained significantly fewer mitotic cells with Ki67 staining. We demonstrated that 22A31 exhibited a growth inhibitory property on HCT116. Our results implied that ERC/mesothelin-targeted therapy might be a promising treatment for colorectal cancer.

## 1. Introduction

Colorectal cancer is a common and lethal disease. According to global cancer statistics (GLOBOCAN estimates), colorectal cancer represented 10.0% of cancer incidence and caused 9.4% of tumor-induced deaths, ranking third by incidence and second among oncological causes of mortality in 2020 [1]. Although great efforts have been made to treat this disease, new therapeutic strategies are still required to improve the prognosis.

Expressed in Renal Carcinoma (ERC)/mesothelin is a glycosylphosphatidylinositol-anchored cell surface protein that is expressed in normal human mesothelium [2]. It is also overexpressed in many malignant tissues, such as mesothelioma [2], pancreatic cancer [3], ovarian cancer [4], lung adenocarcinoma [5], uterine serous carcinoma [6], acute myeloid leukemia [7], and cholangiocarcinoma [8]. ERC/mesothelin is localized on the cellular membrane in these tissues.

As for the molecular function of ERC/mesothelin, the overexpression experiments showed that it stimulates cellular proliferation by enhancing signaling pathways such as mitogen-activated protein kinase/ERK, phosphoinositide 3-kinase/Akt, and nuclear factor-κB/interleukin-6/Stat3 [9,10,11]. However, the factors connecting ERC/mesothelin and these signal pathways are not yet fully understood. Another function of ERC/mesothelin is that it binds to CA125/MUC16 [12]. The binding of ERC/mesothelin to CA125/MUC16 enhances the migration and invasion of carcinoma cells [12,13].

ERC/mesothelin-targeting therapies have been developed to treat malignancies. Amatuximab (MORAb-009), a monoclonal anti-ERC/mesothelin antibody, inhibits the interaction of ERC/mesothelin with CA125/MUC16 [14,15,16] and reduces the cancer’s ability to invade other tissues. Amatuximab also causes cell death by antibody-dependent cell-mediated cytotoxicity (ADCC) [14]. Recently, the modified anti-ERC/mesothelin therapies have been developed to enhance its anti-tumor activities [11,17], and these new agents include antibodies linked to Pseudomonas exotoxin [18], or those conjugated with a tubulin inhibitor [19], or anti-ERC/mesothelin CAR-T cells [20,21,22]. Up to now, anti-ERC/mesothelin therapies have been tried to treat ovarian cancer, pancreatic cancer, mesothelioma, and hematological malignancies [11], but not colorectal cancer.

Recent studies revealed that ERC/mesothelin is expressed in about 50–60% of colorectal adenocarcinoma [23,24,25] but not in normal colon epithelium [25]. Based on these findings, we assumed ERC/mesothelin-targeting might be beneficial for the treatment of colorectal cancer. We previously reported that 22A31, an antibody for the C-terminal part of ERC, suppressed mesothelioma growth via antibody-dependent cell-mediated cytotoxicity (ADCC) in a mouse xenograft model [26]. In this study, we examined whether 22A31 also inhibits colorectal cancer growth or not, also using a mouse xenograft model. We showed that 22A31 treatment induced the growth inhibition of colorectal adenocarcinoma cells by suppressing the proliferating cellular activities.

## 2. Materials and Methods

### 2.1. Mice

Ten 7- to 8-week-old female BALB/c athymic nude (BALB/c nu/nu) mice were purchased from Oriental Yeast Co. Ltd (Tokyo, Japan). All mice in this study were kept under specific pathogen-free conditions in a temperature- and humidity-controlled facility (23 ± 1 °C, 55 ± 5% humidity) on a 12-h light/dark cycle. Water and food were freely available. We performed all in vivo studies according to the guidelines of the Laboratory Animal Experimentation Committee of Juntendo University School of Medicine (Permit Number 260226).

### 2.2. Cells and Antibodies

HCT116 cells (RCB2979) [27] derived from human colorectal adenocarcinoma were obtained from the RIKEN cell bank (Ibaraki, Japan) and cultured in Dulbecco’s modified Eagle’s media supplemented with 10% fetal bovine serum. SW1116 (CCL-233) and SW480 (CCL-228), established from human colorectal adenocarcinoma, NCI-H226 (CRL-5826) [28], established from human mesothelioma, and Jurkat (TIB-152), established from human T cell leukemia, were provided from the American Type Culture Collection (Manassas, VA, USA), and cultured in RPMI-1640 medium supplemented with 10% fetal calf serum. ERC/mesothelin-specific mouse monoclonal antibody, 22A31, was established in our laboratory as previously described [29]. Normal mouse IgG1κ (MOPC-21, M969) was purchased from Sigma (St. Louis, MO, USA). Anti-human-specific Ki-67 antigen monoclonal antibody (clone MIB-1, M7240) was purchased from Dako (Glostrup, Denmark). Anti-cleaved caspase-3 (Asp175) antibody (#9661) was purchased from Cell Signaling Technology (Danvers, MA, USA). Anti-FAS antibody (CH-11) was purchased from MBL life sciences (Tokyo, Japan).

### 2.3. Flow Cytometric Analysis

ERC/mesothelin expression on the cell surface was analyzed by flow cytometry. Cells (2 × 10^5^) were incubated with 2 μg/mL of 22A31 or 2 μg/mL of normal mouse IgG1κ diluted in phosphate-buffered saline (PBS) containing 1% bovine serum albumin (BSA) (017-23294, Wako pure chemical industries Ltd., Osaka, Japan) (1% BSA in PBS) at 4 °C for 40 min. After washing with 1% BSA in PBS, cells were resuspended in 80 μL of 1% BSA in PBS containing 2 µg/mL of Alexa Flour 488-conjugated goat anti-mouse IgG (Molecular Probes, Eugene, OR, USA) and incubated at 4 °C for 30 min. After washing with 1% BSA in PBS, the stained cells were counted by LSR Fortessa (BD Bioscience, San Jose, CA, USA) and analyzed by FlowJo (version 10.5.3) (BD Bioscience). As for cell cycle analysis, HCT116 cells treated with 22A31 (0.2 and 2.0 μg/mL) or with control IgG1κ (2.0 μg/mL) at 37 °C for 48 h were collected and incubated in a hypotonic buffer (0.3% Nonidet P-40, 50 μg/mL propidium iodide, 0.058% NaCl, 1 U/mL RNase, 0.1% sodium citrate) at 4 °C for 10 min. The percentage of G1, S, G2/M, and sub-G1 populations were analyzed by FACS Celesta (BD Bioscience) and FlowJo software. As for apoptosis analysis, HCT116 cells treated with 22A31 (0.2 and 2.0 μg/mL) or control IgG1κ (2.0 μg/mL) and Jurkat cells treated with anti-FAS antibody (0 and 0.2 μg/mL) at 37 °C for 24 h were collected and suspended in an Annexin-binding buffer (2% BSA, 140 mM NaCl, 2.5 mM CaCl_2_, 10 mM HEPES, pH 7.4). One hundred µL of cell suspension (2 × 10^5^) was incubated with 5 µL of Alexa Fluor 647-conjugated Annexin (A35109) (Invitrogen Japan, Tokyo, Japan) at room temperature for 15 min. Then 400 µL of an Annexin-binding buffer and 3 µL of 50 μg/mL DAPI [4’, 6-diamidino-2-phenylindole] were added to cells, and the mixtures were kept on ice. Annexin-V positive cells were counted by FACS Celesta and analyzed by FlowJo. All experiments were performed three times to verify the results.

### 2.4. Anti-Tumor Activity of 22A31 mAb in BALB⁄c nu⁄nu Mice

HCT116 cells (1 × 10^6^) were injected subcutaneously into the backs of BALB/c nu/nu mice. All mice were randomly allocated to either an antibody treatment group with 22A31 or isotype-matched control IgG1κ. Each treatment group was composed of five mice. There was no difference in body weights between 22A31 group (*n* = 5) and control IgG1κ group (*n* = 5) (20.6 ± 1.4 g vs. 19.4 ± 1.5 g) on the day of HCT116 injection. When the tumor volume exceeded 40 mm^3^, intraperitoneal injection of 22A31 or IgG1κ (200 µg/each body) was started (Day 1) and repeated twice per week for 4 weeks. Subcutaneous tumors were measured by calipers every day, and tumor volume was calculated using following formula: V = [π/6] × a × b × c (a: length of major axis, b: length of minor axis, c: height). In case of multi-nodular tumors, the volume was calculated as a sum of each spherical mass.

On Day 29, mice were euthanized by 100% CO_2_ at a flow rate of 20% of the chamber volume per minute. They were kept in 100% CO_2_ for 10 min to ensure that they were dead. Their euthanasia was confirmed by the absence of cardiac beats and respiratory movements. Subcutaneous tumors, lungs, and livers were sectioned with 2 mm thickness, and all samples were fixed in 10% buffered formalin at room temperature overnight, processed for paraffin-embedding, and then sectioned with 3 μm thickness. For hematoxylin-eosin (HE) staining, deparaffinized sections were stained in hematoxylin for 3 min and in eosin for 2 min at room temperature. All histological analysis was performed by a pathologist (Kazunori Kajino, Department of Human Pathology, Juntendo University School of Medicine, Tokyo, Japan).

### 2.5. Immunohistochemistry

After deparaffinization, tissue sections were heated in 10 mM citrate buffer (pH 6) for antigen retrieval and then treated with 3% hydrogen peroxide. The sections were incubated with anti-Ki-67 (1:100) or anti-cleaved caspase-3 (1:200) antibodies diluted in Dako REALTM Antibody Diluent (S2022) at 4 °C overnight. After three washes with Tris Buffered Saline-Tween20, the sections were incubated with Dako EnVisionTM+ System-horse radish peroxidase for 1 h. Diaminobenzidine was used as the substrate for peroxidase. As a positive control of cleaved caspase-3 staining, colorectal carcinoma tissue was used because colorectal carcinoma is reported to be positive for cleaved caspase 3 [30].

### 2.6. TdT-Mediated dUTP Nick End Labeling (TUNEL) Assay

The detection of apoptosis in the tumor tissues was performed using the DeadEnd^TM^ Colorimetric TUNEL System (G7130, Promega, Madison, WI, USA) following the manufacturer’s protocols. As positive controls of TUNEL assay, all specimens were treated by 10 unit/mL of DNase I for 10 min at room temperature before TUNEL procedures. This treatment made more than 80% of nucleus positive in all specimens, and one representative picture is shown as a positive control.

### 2.7. Quantification

The number of mitotic cells in tumors was counted in fields of view at 400× in HE-stained specimens. In order to calculate the Ki-67 labeling index, the number of Ki-67 positive and whole cells in a field of view at 400× were counted. The following formula was used for calculations: Ki-67 labeling index = Ki-67 positive cell count/whole-cell count × 100 (%). The numbers of cleaved caspase 3-positive cells or TUNEL positive cells in each tumor were counted in a given field of view at 200× or at 400×, respectively. The positive signals of cleaved caspase 3 were distributed more unevenly than those of Ki-67, TUNEL, and mitotic cells. Therefore, we counted cleaved caspase 3-positive signals in 200× fields and other three in 400×. In all analyses of quantification, 10 randomly selected fields of view per tumor were subjected to a cell count, and the average of 10 fields was calculated for each mouse. Numbers of mitotic cells, Ki-67, TUNEL, and cleaved caspase 3-positive signals were counted by a pathologist (Kazunori Kajino, Department of Human Pathology, Juntendo University School of Medicine, Tokyo, Japan).

### 2.8. Statistical Analysis

A Mann–Whitney U test was used to compare the cell staining rate by immunohistochemistry, mitotic cell rate, and tumor volume between 22A31-treated and control groups. All statistical analyses were performed using SPSS version 19 (SPSS, Chicago, IL, USA). The significance level for statistical testing was set at *p* < 0.05.

## 3. Results

### 3.1. ERC⁄Mesothelin Expression in Colorectal Adenocarcinoma Cells

Figure 1 shows that ERC/mesothelin was expressed in all three colorectal adenocarcinoma cell lines, HCT116, SW1116, and SW480. We selected HCT116 for our further experiments because a peak of ERC/mesothelin-positive cells (red line) and that of control cells (black dotted line) were more clearly separated in HCT116 than in SW1116 or SW480.

### 3.2. Anti-Tumor Activity of 22A31 in Mice Xenograft Model

Our experimental protocol is shown in Figure 2A. The average times needed for the tumor volumes of both 22A31- and IgG1κ-treated groups to reach 40 mm^3^ were 9.6 and 9.5 days, respectively. After treatments were started, one mouse in the IgG1κ group whose tumor growth was markedly fast died on Day 20. Five mice in the 22A31-treated group and four mice in the IgG1κ-treated group completed the protocol (Figure 2A). Tumor growth was apparently inhibited in the 22A31-treated group after the start of treatment compared to the IgG1κ-treated group. At the end of the protocol on Day 29, tumor volumes of mice in the 22A31-treated group were significantly smaller than those of mice of the IgG1κ-treated group (Figure 2B). Collectively, these findings showed that 22A31 suppressed the growth of HCT116 xenograft tumors. All the raw data corresponding to Figure 2B are shown in Appendix A. We also investigated whether liver or lung metastases coexisted at the time of sacrifice. Although no liver metastasis was observed, we detected lung micro-metastases harbored by three out of five mice in the 22A31-treated group and two out of four mice in the IgG1κ-treated group. There was no significant difference in lung metastases between the two groups.

### 3.3. Characteristics of Xenograft Tumor Cells

In this study, we showed the inhibitory effect of 22A31 on the growth of primary tumors of subcutaneously colorectal adenocarcinoma xenografts. To investigate whether 22A31 changed any biological characteristics of tumors, we investigated cell proliferation and death rates in such tumors. First, we counted the number of mitotic cells in xenograft tumors; tumors from mice in the IgG1κ-treated group contained significantly more mitotic cells compared to those of mice in the 22A31-treated group (*p* < 0.05; Figure 3A–C). Moreover, immunohistochemistry showed a significantly higher Ki-67 labeling index in tissues of mice in the IgG1κ-treated group than those of the 22A31-treated group (*p* < 0.05; Figure 3D–F). Then we tried to assess the direct effect of 22A31 on HCT116 proliferation in vitro by flow cytometry. We did not find any changes in cell cycles of HCT116 treated with 22A31 (Figure 4). These findings suggested that the effect of 22A31 to suppress cell proliferation was seen only in vivo and not in vitro.

We subsequently performed immunohistochemistry with an anti-cleaved caspase 3 antibody as the primary antibody. This experiment did not show a significant difference in positive cleaved caspase 3 staining between the two groups (Figure 3G–J). We also performed a TUNEL assay, and it did not show a significant difference in the numbers of TUNEL-positive cells between the two groups (Figure 3K–N). 

Additionally, we performed flow cytometric analyses to assess the induction of apoptosis in HCT116 cells by 22A31 in vitro. After 24 h incubation with several different concentrations of 22A31 treatment, HCT116 cells were subjected to flow cytometric analysis to detect Annexin-V positive cells. The 22A31 treatment did not cause any changes in Annexin-V positivity of HCT116 cells (Figure 5). All these results showed that the 22A31 treatment did not cause significant apoptosis of HCT116 both in vivo and in vitro.

## 4. Discussion

In this study, we described an inhibitory effect of the 22A31 antibody on tumor growth in a colorectal cancer xenograft mouse model. It was unexpected that 22A31 suppressed tumor growth without evidence of cell death. We previously showed that the anti-tumor effect of 22A31 on mesothelioma cells was mediated by antibody-dependent cell-mediated cytotoxicity (ADCC) [26] associated with perforin and granzyme to damage tumor cells [31]. Therefore, at first, we expected 22A31 to induce increased apoptosis in tumors compared to IgG1κ. Immunohistochemistry of cleaved caspase 3 and TUNEL assay, however, failed to show apoptosis induction by 22A31 (Figure 3G,K). We also showed that 22A31 did not cause apoptosis of HCT116 in vitro by the measurement of Annexin-V positive cells in flow cytometric analyses (Figure 5). We cannot deny the involvement of ADCC in the inhibition of tumor growth in this study. Therapy with 22A31 was undertaken for 4 weeks. Even if ADCC did occur, it was possible that tumor cell death caused by ADCC had occurred and was completed much earlier than when mice were sacrificed and thus remained undetected. We also admit that a single experiment of five mice per group is not enough to draw rigorous conclusions about the involvement of apoptosis in our xenograft systems. To clarify these controversies, further study that assesses the early response to 22A31 therapy in orthotopic models should be conducted using a large number of mice per group. An anti-proliferative tendency was, however, observed at the time of assessment even after 4 weeks of 22A31 therapy.

Recently, Inoue et al. presented the data showing that overexpression of ERC/mesothelin enhances the proliferative activity of HCT116 [25]. Other groups showed that its overexpression in cultured cells stimulates cellular proliferative activities by enhancing signaling pathways such as mitogen-activated protein kinase/ERK, phosphoinositide 3-kinase/Akt, and nuclear factor-κB/interleukin-6/Stat3 [9,10]. As expected, inhibition of ERC/mesothelin expression suppresses the cellular proliferative activity [32]. All these data indicate that ERC/mesothelin has the function of stimulating cellular proliferation. The question raised in our study is “How can extracellular 22A31 suppress cell proliferation?” We speculate several possible mechanisms. First, ERC/mesothelin localizes on the membrane of normal mesothelium and malignant cells both in vivo and in vitro, and these findings suggest that ERC/mesothelin exerts its function on the cellular membrane. The binding of the extracellular 22A31 to ERC/mesothelin on the membrane possibly inhibits its function of stimulating cellular proliferation. ERC/mesothelin may work as a receptor for some ligands. Although GPI-anchor proteins lack intracellular domains, some of them can associate with protein tyrosine kinases and transduce signals [33]. This hypothesis is, however, not compatible with the findings in Figure 4 showing that 22A31 in vitro did not influence the cell cycles of HCT116, and this result suggested us the involvement of some in vivo factors may be required for 22A31 to exert anti-proliferative activities. If we can assume ERC/mesothelin as a receptor to stimulate signal transduction, the effectiveness of 22A31 is dependent on the amount or character of ligands. It is possible that such effective ligands exist only in vivo and not in vitro. Second, Rump et al. [12] demonstrated ERC/mesothelin binds to CA125 (also named MUC16) and that the co-expression of ERC/mesothelin and CA125/MUC16 is observed in high-grade ovarian adenocarcinoma. Likewise, Shimizu et al. [13] showed that the co-expression of ERC/mesothelin and CA125/MUC16 enhances migration and invasion of pancreatic ductal adenocarcinoma and is related to a poor prognosis. It is possible that 22A31 interrupts the heterotypic interaction between ERC/mesothelin and other molecules such as CA125/MUC16 on the surface of tumor cells or adjacent stromal cells, leading to the suppression of proliferative activities of carcinoma cells. Further studies are required to validate these hypotheses.

Another unexpected result was that the intraperitoneal injection of 22A31 did not have a suppressive effect on metastatic lesions caused by subcutaneous colorectal adenocarcinoma xenografts despite 22A31 inhibiting tumor growth. Our study failed to confirm a suppressive effect on metastasis by 22A31. However, the number of metastases obtained in this study was too few to be analyzed statistically. To investigate the effect on metastasis, we must design another animal study model that enables us to observe much more metastases, especially liver metastases that are the most frequent metastases in colorectal cancer.

## 5. Conclusions

Clinical trials of ERC/mesothelin-targeting agents have confirmed their safety and utility in mesothelioma and ovarian and pancreatic cancers. Our results in this study revealed that 22A31 inhibited the growth of colorectal adenocarcinoma cells. Further studies are required for us to clarify whether ERC/mesothelin works as a receptor or a ligand of signal transduction pathways by searching the binding proteins of ERC/mesothelin other than CA125/MUC16.

## Figures and Tables

**Figure 1 cancers-14-02198-f001:**
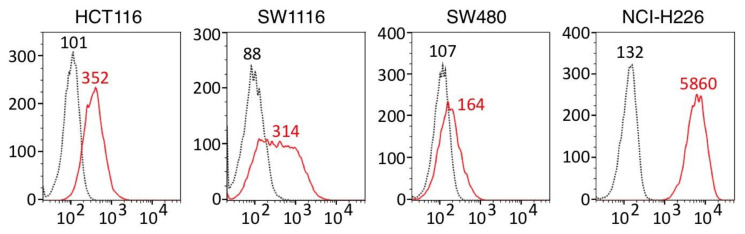
Cell surface ERC/mesothelin expression on human colorectal adenocarcinoma cells. Membranous ERC/mesothelin expression was measured by flow cytometry in human colorectal adenocarcinoma cell lines (HCT116, SW1116 and SW480). Cells reacting with anti-ERC/Mesothelin (22A31) or mouse IgG1κ (isotype-matched control) are indicated by red solid lines or black dashed lines, respectively. H226, NCI-H226 human mesothelioma cell used as a positive control. Median fluorescence intensity of 22A31- or IgG1κ-treated cells are shown in red or black letters respectively in each histogram.

**Figure 2 cancers-14-02198-f002:**
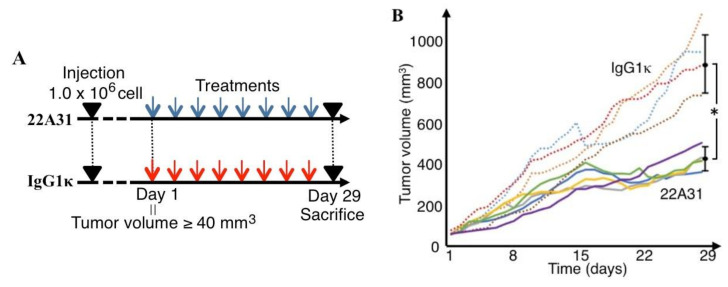
Anti-tumor activity of 22A31 in HCT116 human colorectal adenocarcinoma cells. (**A**) Experimental protocol. Day 1 was defined as when the tumor volume was over 40 mm^3^. All mice were treated eight times with 200 µg 22A31 or IgG1κ (twice per week), and euthanized on Day 29. (**B**) Tumor volumes in 22A31 and IgG1κ treatment groups. Solid lines and dashed lines represent the tumor volumes of 22A31 and IgG1κ treatment groups respectively. The average tumor volume ± SD in each of 22A31 and IgG1κ treatment groups on day 29 are shown. * *p* < 0.01.

**Figure 3 cancers-14-02198-f003:**
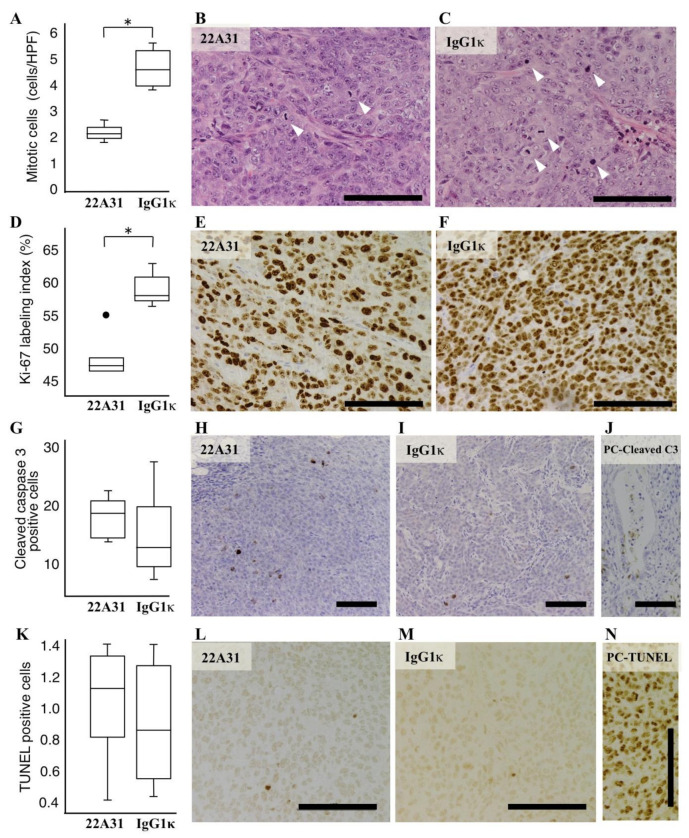
Characteristic difference in HCT116 human colorectal adenocarcinoma tumors between 22A31 and IgG1κ treatment groups. (**A**) The average mitotic cell number in a field of view at 400× magnification. * *p* < 0.05. (**B**,**C**) Representative images of hematoxylin and eosin staining in tissues from mice in 22A31 and IgG1κ treatment groups. Mitotic cells are indicated by white arrowheads. (**D**) Ki67 labeling index at 400× magnification. * *p* < 0.05. (**E**,**F**) Representative images of Ki67 staining in 22A31 and IgG1κ groups. (**G**) The average cell numbers in a field of view showing positive cleaved caspase 3 staining at 200× magnification. (**H**,**I**) Representative images of cleaved caspase 3 staining in tissues from mice in 22A31 and IgG1κ treatment groups. (**J**) Colon carcinoma tissue was used as a positive control of cleaved caspase 3 staining [30] (**K**) The average cell numbers in a field of view showing TUNEL positive cells at 400× magnification. (**L**,**M**) Representative images of TUNEL assay in tissues from mice in 22A31 and IgG1κ treatment groups. (**N**) A positive control of TUNEL staining was prepared by DNase I treatment of specimens before TUNEL assay. Scale bars, 100 µm in all figures.

**Figure 4 cancers-14-02198-f004:**
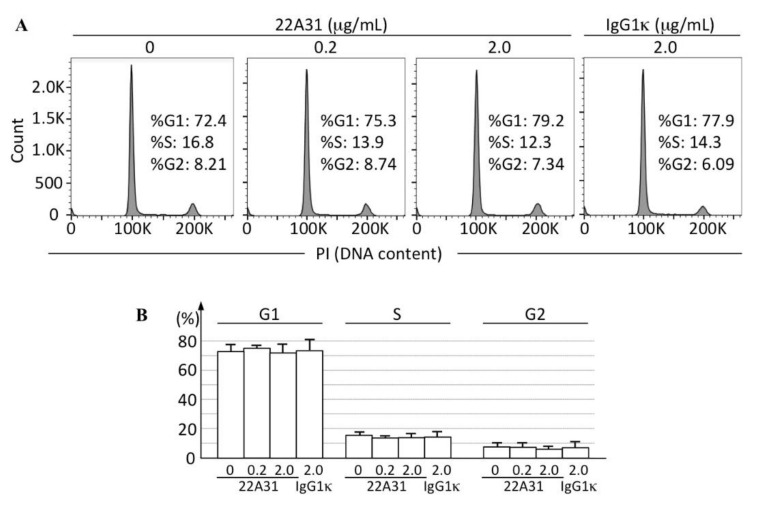
Cell cycle analyses of HCT116 cells after treatment with 22A31. The percentage of G1, S, and G2 populations were analyzed by flow cytometry. (**A**) Representative histograms. (**B**) Quantified data of G1, S, and G2 populations (mean ± SD) in three experiments. HCT116 cells were incubated with 0, 0.2 and 2.0 μg/mL 22A31 at 37 °C for 48 h. As a negative control, 2.0 μg/mL IgG1κ was used.

**Figure 5 cancers-14-02198-f005:**
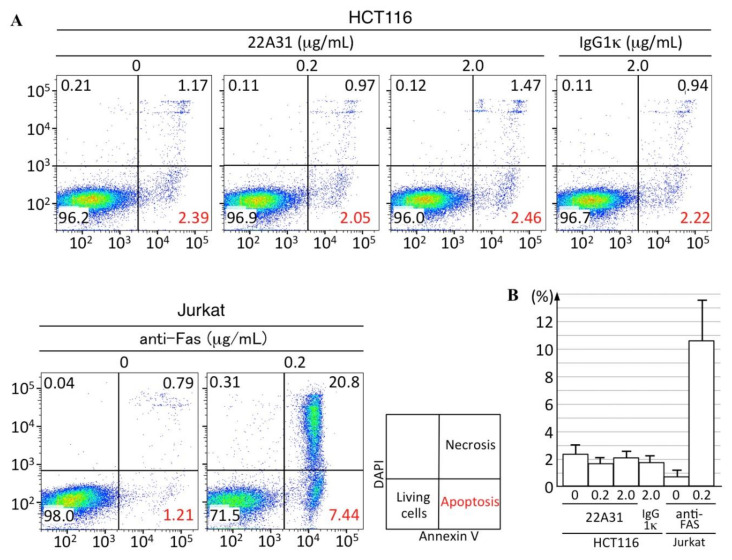
Apoptotic cells in HCT116 detected by Annexin V positivity after treatment with 22A31. (**A**) Representative dot-plots. The percentage of apoptotic cells is shown in red numbers in a right lower quadrant of each dot-plot. (**B**) Quantified data of apoptotic cells (mean ± SD) in three experiments. HCT116 cells were incubated with 0 or 0.2 or 2.0 μg/mL 22A31 at 37 °C for 24 h and then were subjected to flow cytometric analysis. As a negative control, HCT116 cells were treated with IgG1κ 2.0 μg/mL. As a positive control, Jurkat cells were treated with anti-FAS 0 or 0.2 μg/mL.

## Data Availability

The data presented in this study are available in the article and Appendix A.

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
