# Peer review of "The Inhibitory Effects of Anti-ERC/Mesothelin Antibody 22A31 on Colorectal Adenocarcinoma Cells, within a Mouse Xenograft Model"

_cancers, 2022, doi:10.3390/cancers14092198_

Round 1

Reviewer 1 Report

Overall it is a good paper and experimentally sound.  it is well written and easy to understand. Good referencing as well.   The authors created a monoclonal antibody against MSLN (22A31, publication in 2009), then tested it in vivo first in a mesothelioma mouse model (publication in 2010) showing 22A31 works well and reduces tumour growth via NK cell-mediated ADCC. Here they show similar results in a mouse model of colorectal adenocarcinoma (subcutaneous tumours in immunodeficient Balb/c nude mice).

They show 22A31 works to reduce tumour growth, but they concede that the precise mechanism of action in colorectal adenocarcinoma is unclear and requires further study.  

In order for this to be convincing ADCC assays should be included in this paper to confirm NK-dependent ADCC in colorectal adenocarcinoma (similarly to their previous paper in 2010 in mesothelioma mouse model).

  ·         No effect seen on cell proliferation in vitro (shown also previously in the paper in 2010 in mesothelioma cell lines), only in vivo. -> suggests cell proliferation inhibition is indirect -> well discussed in the discussion section

·         Figure 1 shows ERC/MSLN expression via mean fluorescent intensity. Both mean and median are accepted units of measures, however median fluorescent intensity is considered a more precise unit of measure as it is less sensitive to outliers. This should be addressed

·         Apoptosis analysis: they only have a positive control using Fas-antibody in Jurkat cells but not HCT116 cells, why not use fas-antibody in HTC116 cells as positive control? Should be included.

·         In the group’s previous paper (2010, reference n. 15) they showed how cytotoxicity was dependent on NK cell-mediated ADCC using mesothelioma cell lines -> the authors should repeat these ADCC assays with colorectal adenocarcinoma cell lines and in vivo

Author Response

Re: cancers-1649806

We appreciate kind and supportive comments by three (two in the first revise, and one in the second revise) reviewers to improve our manuscript.  We tried to respond to these comments as much as possible.

The reviewers’ comments are shown in blue letters and our responses are shown in red letters.

Our responses to comments by the third reviewer (in the second revise)

Comments: Overall it is a good paper and experimentally sound.  it is well written and easy to understand. Good referencing as well.   The authors created a monoclonal antibody against MSLN (22A31, publication in 2009), then tested it in vivo first in a mesothelioma mouse model (publication in 2010) showing 22A31 works well and reduces tumour growth via NK cell-mediated ADCC. Here they show similar results in a mouse model of colorectal adenocarcinoma (subcutaneous tumours in immunodeficient Balb/c nude mice).

They show 22A31 works to reduce tumour growth, but they concede that the precise mechanism of action in colorectal adenocarcinoma is unclear and requires further study.  

Our responses: As mentioned by the reviewer, further study is required for us to clarify how 22A31 suppresses the proliferative activity of carcinoma cells.

Comments: In order for this to be convincing ADCC assays should be included in this paper to confirm NK-dependent ADCC in colorectal adenocarcinoma (similarly to their previous paper in 2010 in mesothelioma mouse model).

Our responses: We did not perform ADCC assays in this study and were not able to add the data during the limited time for revise. The characteristic finding of our study is that 22A31 suppressed the tumor growth, by the suppression of proliferative activity of carcinoma cells, regardless whether ADCC was involved or not. We cannot deny nor support the involvement of ADCC. Further study is required for us to clarify how 22A31 suppresses the proliferative activity of carcinoma cells.

Comments: No effect seen on cell proliferation in vitro (shown also previously in the paper in 2010 in mesothelioma cell lines), only in vivo. -> suggests cell proliferation inhibition is indirect -> well discussed in the discussion section

Our responses: 22A31 seems to inhibit the interaction of ERC/mesothelin and some in vivo factors.

Comments: Figure 1 shows ERC/MSLN expression via mean fluorescent intensity. Both mean and median are accepted units of measures, however median fluorescent intensity is considered a more precise unit of measure as it is less sensitive to outliers. This should be addressed

Our responses: Following the reviewer's comments, we replaced the numbers of "mean fluorescent intensity" with those of "median fluorescent intensity" in Figure 1. We also changed the corresponding description in the legend of Figure 1.

Comments: Apoptosis analysis: they only have a positive control using Fas-antibody in Jurkat cells but not HCT116 cells, why not use fas-antibody in HTC116 cells as positive control? Should be included.

Our responses: We were not able to perform the apoptosis analysis using fas-antibody and HCT116 cells during the limited time for revise. The other reviewer, in the first revise step, suggested us to add the data of untreated Jurkat cells as a negative control.  We had performed the apoptosis analysis of HCT116 treated by 0 mg/mL anti-Fas, and we added the data in Figure 5.

Comments: In the group’s previous paper (2010, reference n. 15) they showed how cytotoxicity was dependent on NK cell-mediated ADCC using mesothelioma cell lines -> the authors should repeat these ADCC assays with colorectal adenocarcinoma cell lines and in vivo

Our responses: We are sorry but we have not performed ADCC assay and we cannot perform it in the limited time.  Please see our response mentioned above.

Reviewer 2 Report

In this manuscript, Taniguchi and collagues have investigated the inhibitory effects of anti-ERC Ab 22A31 on colorectal adenocarcinoma cells HCT116, within a mouse xenograft model.

The study is clearly described and properly designed. The materials employed and the practical approach are illustrated in detail and the results are consolidated by explicit figures, yet the introduction could be improved.

Minor comments: I suggest to double check the document, since a few simbols are often not correctly reported, i.e. °, k and μ. Furthermore, the structure of the phrase in lines 260-262 could be more clear and may i suggest to simplify the title of the essay.

Author Response

Re: cancers-1649806

 We appreciate kind and supportive comments by two reviewers to improve our manuscript. We tried to respond to these comments as much as possible.

The reviewers comments are shown in blue letters and our responses are shown in red letters.

Our responses to Reviewer 1's comments

Comments: In this manuscript, Taniguchi and colleagues have investigated the inhibitory effects of anti-ERC Ab 22A31 on colorectal adenocarcinoma cells HCT116, within a mouse xenograft model.

Our responses: We are impressed by the reviewer's comment that simply summarizes our work. This reviewer suggested us to simplify the title of our paper in later comments. Following the comment shown above, we changed the title to "The inhibitory effects of anti-ERC/mesothelin antibody 22A31 on colorectal adenocarcinoma cells HCT116, within a mouse xenograft model."

Comments: The study is clearly described and properly designed. The materials employed and the practical approach are illustrated in detail and the results are consolidated by explicit figures, yet the introduction could be improved.

Our responses: We changed Introduction so that it is more clearly associated with Results and Discussion.

Comments: I suggest to double check the document, since a few symbols are often not correctly reported, i.e. °, k and μ.

Our responses: We are very sorry that °, k and μ were incorrectly shown. This is mostly because of our failure, but we are afraid that few of them are changed during the uploading steps. We carefully checked and corrected them.

Comments: Furthermore, the structure of the phrase in lines 260-262 could be more clear and may i suggest to simplify the title of the essay.

Our responses: The phrase in lines 260-262 are changed to be a simple sentence "It was unexpected that 22A31 suppressed tumor growth without evidence of cell death." in Discussion. Title was changed as mentioned above.

Reviewer 3 Report

The authors are to be congratulated in their exploration of mesothelin (MSLN) antibody inhibition of colon cancer cell proliferation. Overall, the work proceeded in a logical fashion. The manuscript could be strengthened with the following changes:

  1. The introduction should be expanded to better contextualize the results that follow. For example, a description of what is known about MSLN signaling (e.g., what is known about potential ligands and downstream signaling pathways), the presumed mechanism of action for the anti-MSLN antibodies in other cancers, and the known expression levels in normal colonic tissue should be added.
  2. References should be as up to date as possible. The first reference is describing disease incidence and mortality but is a decade old. 
  3. The methods could be presented with more clarity. For example, there is no description of who quantified the Ki67 cells (were they a pathologist? a student? were there more than one person making the assessments and inter-user variability assessed?). Furthermore, there is no description of why the decision to count at 400x for KI67 and 200x for TUNEL staining was made.  Some results are presented that do not have a description of methods (e.g., how were micrometastases in the lungs detected? did they submit the entire lungs for histology? did a pathologist examine these? or was it done by flow cytometry?)
  4. Results: Figure 2B is unclear in that the manuscript states 5 mice per group, but I cannot appreciate 5 separate lines in red and in blue? Addition of individual tumor pictures after sacrifice could be incorporated, or, perhaps even better, a table of individual tumor volumes by day. Since staining was so modest for CC3 and TUNEL as shown in Figure 3H,I,K and L, a positive control for staining should be added. In Figure 3B and C, the background staining is much darker in C so if there is another sample slide for C that would be better to include. In addition, examples of mitotic figures as were counted should be shown by arrows in Figures B and C. As anti-FAS treatment of Jurkat cells was used as a positive control for the authors' apoptosis staining and gating strategy, a non-treated Jurkat sample dot plot would be a nice addition for completeness sake.
  5. Conclusions: The authors could offer additional explanations as to WHY they think the antibody decreased proliferation in vivo but did not have the expected effects on apoptosis. Perhaps this was a study power issue, and assessment of effect size for apoptosis as an endpoint from the literature would be helpful. In addition, perhaps the mechanism of action in vivo is antibody competition for binding with ligand (MUC16) at the receptor site, thereby decreasing signaling. If I understand the authors in vitro work correctly, there was no ligand available for the added antibody to compete with for the receptor so perhaps this is why there was no apoptosis observed with or without the antibody in vitro? If this is the case, perhaps the authors could propose future studies examining the effects of ligand/receptor binding in CRC and determine if proliferation effects are more pronounced than apoptosis inhibitory effects in CRC lines? 
  6. Since no immune endpoints were examined, I recommend the authors decrease emphasis on this in their discussion section. 

The project could be strengthened by considering the following:

  1. Use of only female mice requires justification and a single experiment of 5 mice per group is too small to draw rigorous conclusions.
  2. Use of subcutaneous models does not recapitulate the appropriate anatomic context for a tumor microenvironment and going forward, use of orthotopic models would be superior. In addition, more than one cell line or model would be of benefit to better be able to generalize findings. 

Author Response

Re: cancers-1649806

We appreciate kind and supportive comments by two reviewers to improve our manuscript. We tried to respond to these comments as much as possible.

The reviewers comments are shown in blue letters and our responses are shown in red letters.

Our responses to Reviewer 2's comments

Comment 1: The introduction should be expanded to better contextualize the results that follow. For example, a description of what is known about MSLN signaling (e.g., what is known about potential ligands and downstream signaling pathways), the presumed mechanism of action for the anti-MSLN antibodies in other cancers, and the known expression levels in normal colonic tissue should be added.

Our responses: We reconstructed Introduction, so that it is more clearly associated with Results and Discussion. We mentioned the following things by citing references.

  1. What is known and what is unknown, as for the molecular function of ERC/mesothelin.
  2. Reported activities of Amatuximab (MORAb-009) on ERC/mesothelin-expressing tumor.
  3. The expression of ERC/mesothelin on colorectal cancer but not in normal colon epithelium.

Comment 2: References should be as up to date as possible. The first reference is describing disease incidence and mortality but is a decade old. 

Our responses: We replaced our old reference of disease incidence and mortality with a latest one (ref 1).

Comment 3-1: The methods could be presented with more clarity. For example, there is no description of who quantified the Ki67 cells (were they a pathologist? a student? were there more than one person making the assessments and inter-user variability assessed?).

Our responses: A pathologist (KK) counted the positive signal of Ki-67, cleaved caspase 3, and TUNEL in 10 microscopic fields in all samples and calculated the average number of positive signals in 1 field to prepare graphs shown in Figure 3. We added these description in Materials and Methods.

Comment 3-2: Furthermore, there is no description of why the decision to count at 400x for KI67 and 200x for TUNEL staining was made. 

Our responses: The positive signals of cleaved caspase 3 (CC3) were distributed more unevenly than those of Ki-67, TUNEL, and mitotic cells. So we counted CC3 positive signals in 200x fields and other three in 400x. We added this description in Materials and Methods. In Figure legend of Figure 3E and 3E, there were incorrect description about the magnification of picture and we corrected them.

Comment 3-3: Some results are presented that do not have a description of methods (e.g., how were micrometastases in the lungs detected? did they submit the entire lungs for histology? did a pathologist examine these? or was it done by flow cytometry?)

Our responses: Lungs and liver of all mice were sliced to 2mm-thickness and all slices were processed for histological examination. A pathologist (KK) observed and counted the micrometasitasis. We added this description in Materials and Methods.

Comment 4-1: Results: Figure 2B is unclear in that the manuscript states 5 mice per group, but I cannot appreciate 5 separate lines in red and in blue?

Our responses: We replaced it with a clear figure in which states of 5 mice are discernible.

Comment 4-2: Addition of individual tumor pictures after sacrifice could be incorporated, or, perhaps even better, a table of individual tumor volumes by day.

Our responses: A table of individual tumor volumes by day is shown in a supplementary Table (Table S1: Size of tumors in control (IgG1k) or 22A31-treated groups on Day 1 to 29.) that was uploaded in the initial submission.

Comment 4-3: Since staining was so modest for CC3 and TUNEL as shown in Figure 3H,I,K and L, a positive control for staining should be added.

Our responses: As for a positive control of CC3 staining, we used colorectal carcinoma tissue (Figure 3J), because it is reported to be positive for CC3 (ref 30). As for a positive control of TUNEL, the all samples are treated by DNase I following the manufacturer's protocol, before TUNEL staining. All samples showed the very similar findings, with more than 80 % of cells positive for TUNEL in all specimen. So we randomly selected the specimen to be presented as a positive control (Figure 3N)

Comment 4-4: In Figure 3B and C, the background staining is much darker in C so if there is another sample slide for C that would be better to include. In addition, examples of mitotic figures as were counted should be shown by arrows in Figures B and C.

Our responses: Both of 3B and 3C were changed to the new ones in which the mitotic cells are more clearly observed. In each picture, we added white arrowheads to indicate the mitotic cells.

Comment 4-5: As anti-FAS treatment of Jurkat cells was used as a positive control for the authors' apoptosis staining and gating strategy, a non-treated Jurkat sample dot plot would be a nice addition for completeness sake.

Our responses: FACS result of non-treated Jurkat sample is added on Figures 5A and 5B.

Comment 5. Conclusions: The authors could offer additional explanations as to WHY they think the antibody decreased proliferation in vivo but did not have the expected effects on apoptosis. Perhaps this was a study power issue, and assessment of effect size for apoptosis as an endpoint from the literature would be helpful. In addition, perhaps the mechanism of action in vivo is antibody competition for binding with ligand (MUC16) at the receptor site, thereby decreasing signaling. If I understand the authors in vitro work correctly, there was no ligand available for the added antibody to compete with for the receptor so perhaps this is why there was no apoptosis observed with or without the antibody in vitro? If this is the case, perhaps the authors could propose future studies examining the effects of ligand/receptor binding in CRC and determine if proliferation effects are more pronounced than apoptosis inhibitory effects in CRC lines? 

Our responses: The reviewer analysed and dissected the result of our study more deeply than we did. Following the comments, we did our best to improve Discussion.

  1. Since no immune endpoints were examined, I recommend the authors decrease emphasis on this in their discussion section. 

Our responses: We tried to reduce the emphasis on immune endopoint.

The project could be strengthened by considering the following:

  1. Use of only female mice requires justification and a single experiment of 5 mice per group is too small to draw rigorous conclusions.
  2. Use of subcutaneous models does not recapitulate the appropriate anatomic context for a tumor microenvironment and going forward, use of orthotopic models would be superior. In addition, more than one cell line or model would be of benefit to better be able to generalize findings. 

Our responses: We completely agree to these comments and try to improve our future experiments, following these comments.

Round 2

Reviewer 1 Report

it would be far superior for a journal like Cancers to seek a more rigourous set of experiments to address the main question as to the mechanism by which the Ab inhibits proliferation. Why was the major revision so urgent?